# Filling and drainage of a subglacial lake beneath the Flade Isblink ice cap, northeast Greenland

Qi Liang[1], Wanxin Xiao[1], Ian Howat[2,3], Xiao Cheng[1], Fengming Hui[1], Zhuoqi Chen[1], Mi Jiang[1], Lei Zheng[1]

[1]School of Geospatial Engineering and Science, Sun Yat-sen University & Southern Marine Science and Engineering Guangdong Laboratory (Zhuhai), Zhuhai, Guangdong, China
[2]Byrd Polar and Climate Research Center, Columbus, OH, USA
[3]School of Earth Sciences, Ohio State University, Columbus, OH, USA

*Correspondence to*: Lei Zheng (zhenglei6@mail.sysu.edu.cn)

**Abstract.** The generation, transport, storage and drainage of meltwater play important roles in the Greenland Ice Sheet (GrIS) subglacial system. Active subglacial lakes, common features in Antarctica, have recently been detected beneath the GrIS and may impact ice sheet hydrology. Despite their potential importance, few repeat subglacial lake filling and drainage events have been identified in Greenland. Here we examine the surface elevation change of a collapse basin at the Flade Isblink ice cap, northeast Greenland, which formed due to sudden subglacial lake drainage in 2011. We estimate the subglacial lake

volume evolution using multi-temporal ArcticDEM data and ICESat-2 altimetry data acquired between 2012 and 2021. Our long-term observations show that the subglacial lake was continuously filled by surface meltwater, with basin surface rising by up to 55 m during 2012-2021 and we estimate $138.2\times10^6$ m³ of meltwater was transported into the subglacial lake between 2012 and 2017. A second rapid drainage event occurred in late August 2019, which induced an abrupt ice dynamic response. We find that the 2019 drainage released much less water than the 2011 event and conclude that multiple factors, such as the volume of water stored in the subglacial lake and bedrock relief, regulate the episodic filling and drainage of the

lake. By comparing the surface meltwater production and the subglacial lake volume change, we find that only ~64% of the surface meltwater descended to the bed, suggesting potential processes such as meltwater refreezing and firn aquifer storage, which need to be further quantified.

## 1 Introduction

The Greenland Ice Sheet (GrIS) has experienced a strong negative mass balance since the 1990s (Shepherd et al., 2020). Mass loss has resulted from a combination of increased dynamic thinning (Enderlin et al., 2014; King et al., 2020) and decreased surface mass balance (SMB) (Fettweis et al., 2017; Noël et al., 2019). Of these, the decline in SMB due to an increase in surface melting and runoff has recently become the dominant contributor (Lenaerts et al., 2019). Moreover, increased runoff may also impact ice sheet dynamics (Hewitt, 2013; van de Wal et al., 2015). Meltwater draining into the

englacial system can be accumulated in crevasses and raise the ice temperature, leading to increases in ice velocities due to

the weaken/soften of the ice sheet (Cavanagh et al., 2017; Liang et al., 2019; Phillips et al., 2013). When meltwater penetrates from the surface to the ice sheet bed, it can lubricate the ice bed interface, reduce basal drag and increase glacier sliding (Joughin et al., 2013; Moon et al., 2014; Zwally et al., 2002). Therefore, the presence and movement of meltwater at the ice bed interface are considered to significantly affect ice dynamics (Meierbachtol et al., 2013). Given the expected

increases in surface meltwater production in a warming climate (Mottram et al., 2017; Sellevold and Vizcaino, 2021), it is of critical importance to understand GrIS hydrology, especially the routing, storage, drainage and recharge of subglacial water. Sixty four subglacial lakes have been identified in Greenland from airborne radio-echo sounding (Bowling et al., 2019; Livingstone et al., 2022). Most of them are stable, showing little or no evidence of volume change or input from the surface, and are located between the Equilibrium Line Altitude (ELA) and the relatively flat, frozen-bedded ice sheet interior. Only a

few hydrologically-active lakes that are recharged by surface meltwater have been identified from ice surface elevation change measurements (Bowling et al., 2019; Howat et al., 2015; Livingstone et al., 2019; Livingstone et al., 2022; Palmer et al., 2015; Willis et al., 2015). Compared to the widely distributed stable subglacial lakes, the active subglacial lakes are affected more directly by surface meltwater and their drainage may significantly influence the glacier flow dynamics (Davison et al., 2020; Livingstone et al., 2019). Despite this importance, our understanding of Greenland's subglacial lakes

has been primarily developed from theoretical studies or inferences from geophysical exploration due to sparsity of direct observations (Davison et al., 2019).

Satellite remote sensing techniques have been used to monitor the subglacial lakes and detect their activities. As an indirect observation of subglacial lake activity, long-term ice surface elevation changes are usually derived from satellite altimetry (e.g., Fricker et al., 2007; Siegfried and Fricker, 2018, 2021). More recently, time-stamped digital elevation models (DEMs)

have been utilized to reveal the detailed patterns of surface deformation (e.g., Livingstone et al., 2019; Willis et al., 2015). A few studies also use the Synthetic Aperture Radar (SAR) speckle tracking (Joughin et al., 2016; Hoffman et al., 2020) and Interferometry SAR (InSAR) (Gray et al., 2005; Neckel et al., 2021) to detect ice surface displacements. However, few studies have investigated the long-term filling and drainage of subglacial lakes in Greenland. In particular, the subglacial lake volume change, water residence times and drainage are still poorly understood.

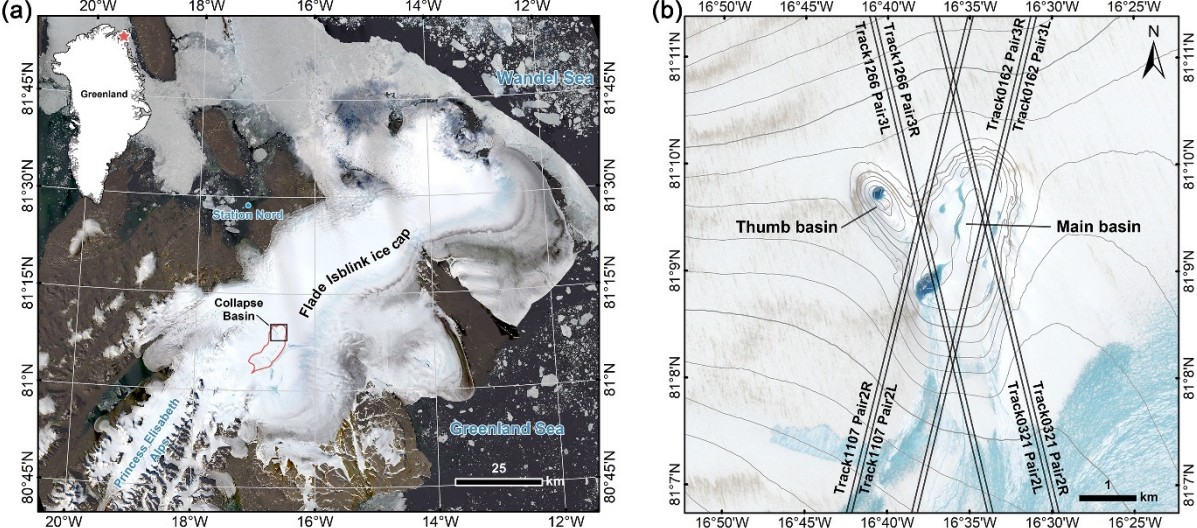

**Figure 1. Map of the study area. (a) Flade Isblink ice cap. Background is a Landsat-8 OLI image acquired on 13 August 2015. The black box shows the location of (b). The red line is the catchment boundary. (b) Sentinel-2 MSI image of the deep basin acquired on 5 August 2020. The grey 10-meter contours are derived from ArcticDEM strips data from 20 April 2015. Black lines indicate the 4 pairs of ICESat-2 single-beam tracks that pass through the collapse basin. The supraglacial meltwater formed in summer usually flows northwards and drains into the ice sheet through crevasses and moulins.**

At the Flade Isblink ice cap (81.3°N, 15.0°W) in northeast Greenland (Figure 1), a collapse basin in the ice cap surface about 70 m deep, created by sudden subglacial lake drainage between August 16 and September 6 in 2011, was first revealed by Willis et al. (2015). Basin surface elevation estimates with DEMs created from stereoscopic satellite imagery suggest that rapid surface uplift occurred over the two years following the collapse, as supraglacial meltwater was transported to the ice base, refilling the subglacial lake. Although Flade Isblink ice cap is not directly connected to the wider GrIS, its glacial setting is similar to that of the northern GrIS. It is important to investigate its behavior and impact on ice dynamics, which may lead to improvements in our understanding of subglacial lakes beneath the GrIS. In order to better understand the repeat subglacial lake filling and drainage, here we extended the surface elevation time series records to early 2021 with ArcticDEM repeat surface models and ICESat-2 altimetry data. We describe the long-term subglacial lake behavior, analyze its volume change and compare it with the surface runoff supply. We also identify a second drainage event in 2019 and explore the impact of drainage on glacier dynamics.

## 2 Data and method

### 2.1 Surface elevation and basin volume change calculation

Surface elevations from 2012 to 2017 are first acquired from multi-temporal ArcticDEM strip data (Porter et al., 2018). The initial absolute accuracy of ArcticDEM strip data is less than 4 meters in horizontal and vertical planes. Therefore, the DEM strips should be vertically co-registered before calculating elevation changes. Only a few DEM strips extend over bedrock or

have ICESat footprints as ground control points in our study area, so we cannot directly co-register each of them. Instead, we first co-register a DEM acquired on 20 April 2015 using the 3-dimensional offset values provided by the metadata text file as a reference. A square window centered over the collapse basin with sides equal to twice the length and width of the basin (~7.6 km), is defined. Another 1500-m buffer is set outward along the boundary of the collapse basin (Figure 2i). Then, all the other DEMs are vertically co-registered to the reference DEM by calculating the mean elevation differences using the pixels within this window but outside the 1500-m buffer. We apply an iterative, 3-standard-deviation filter to remove outliers when estimating the elevation differences (Willis et al., 2015). The DEM precision is estimated from the standard deviation of the elevation differences that remained after the iterative filter. In this way, the influence of both the systematic vertical offsets and snow accumulation or melting are removed.

Besides ArcticDEM data, Advanced Land Observing Satellite (ALOS) Global Digital Surface Model "ALOS World 3D" (AW3D30) (Tadono et al., 2014; Takaku et al., 2014, 2020) is also used to analyze the elevation change. The AW3D30 DEM in our study area is derived from data spanning the period 2006–2010, just before late summer of 2011 when the deep basin formed. As above, the AW3D30 DEM is vertically co-registered to the reference DEM. Note that, the coregistered DEMs only represent 'relative' ice surface heights that have eliminated systematic changes over the larger ice cap due to surface accumulation or melting and other processes, rather than the true elevation.

The surface elevation measurements from the Advanced Topographic Laser Altimeter System (ATLAS) onboard ICESat-2 are also used to extend the time series to early 2021. As a successor to the ICESat-1 satellite mission, ICESat-2, a polar-orbiting satellite with 91-day repeat cycle and 92° orbit inclination, was launched in September 2018 (Markus et al., 2017). ATLAS generates six green (532 nm) laser beams in three pairs along one reference ground track and each pair contains one weak and one strong beam. In across-track direction, the spacing between each beam pair is ~3.3 km and each pair of strong and weak beams are separated by ~90 m. There are 8 tracks (4 pairs) that pass through the collapse basin, with two pairs (Track 0126 pair3 and Track 0321 pair2) passing over the main basin and another two pairs (Track 1266 pair3 and Track 1107 pair2) pass over the area between main basin and thumb basin (Figure 1b). We only used repeat cycles 3-9 for our study because the first two are not repeatss due to pointing control issues.

We use the ICESat-2 level 3a Land Ice Height (ATL06) data product removing poor-quality elevation measurements caused by clouds or random clustering of background photons based on the ATL06 quality summary flag (Smith et al., 2019). Further, we check for height consistency by calculating adjacent elevations using the along-track slope parameter and comparing the estimated to the measured elevations. Only the data where the difference between original elevations and the estimated elevations are less than 2 m are used (Li et al., 2020). In order to reduce errors introduced by large across-track slopes, we merge the two single-beam track data for the left beam and right beam into one beam pair. A reference track is first calculated by averaging all of the single-beam tracks from both left and right ground tracks. The elevation of the reference track for each cycle is then estimated from the left and right single-beam track measurements and the across-track slope parameter (Li et al., 2020). This procedure provides four repeat-track observations for elevation change analysis.

After all of the ICESat-2 data are co-registered to the reference DEM using the method described above, the time series of elevation change over the collapse basin are estimated along the four reference tracks using both the registered ArcticDEM and ICESat-2 data. Additionally, average ice surface elevation changes are also estimated at three reference track crossovers (Figure 2a).

Changes in subglacial water volumes are estimated by integrating elevation change over the basin area. Previous studies
show that a reduction in the depth of the depression would result from the inflow of the ice around the basin (Aðalgeirsdóttir et al., 2000; Willis et al., 2015). Therefore, we expect that the basin volume change here is mainly caused by ice inflow and subglacial lake filling. Assuming the subsidence that occurs around the basin outline in a 1500-m buffer region correspond to ice flowing into the basin, we calculate the inflow volume by integrating the surface elevation changes over the buffer area (Willis et al., 2015). The volume change of the subglacial lake is then estimated by differencing the basin volume change
and ice inflow volume.

## 2.2 Catchment delineation and surface melting analysis

The catchment boundary is extracted using ArcticDEM surface elevation as follows (Smith et al., 2017; Yang et al., 2019). First, we fill the ArcticDEM surface to create a sink-free DEM raster. Then we identify the flow directions from the slope direction on the partially filled DEM. Finally, the Basin function in ArcGIS software is used to delineate the catchment
boundary.

To assess the surface meltwater dynamics, we use estimates of meltwater runoff from the high-resolution Regional Atmospheric Climate Model (RACMO2.3p2) (Noël et al., 2018). Daily runoff produced in the catchment are generated from RACMO2.3p2 that are statistically downscaled to a 1-km horizontal resolution (Noël et al., 2019). The total runoff within the catchment is calculated by summing the 1 km grid cells within the catchment boundary. Furthermore, a series of
Landsat-8 Operational Land Imager (OLI) and Sentinel-2 MultiSpectral Instrument (MSI) images acquired during the 2014-2020 melt season are used to better illustrate the supraglacial lakes and streams.

## 2.3 Ice velocity estimate

We obtain estimates of the ice surface velocity from the MEaSUREs Greenland Monthly Ice Sheet Velocity Mosaics from SAR and Landsat dataset, Version 3 (Joughin et al., 2018). These include monthly surface velocity estimates for the
Greenland Ice Sheet and periphery and are posted at a 200 m grid resolution. Due to the limited coverage of the ice velocity product in the summer, an 800 m by 800 m region located downstream of the collapse basin is chosen to evaluate changes in ice surface velocity (Figure 6c). We calculate the mean velocity within this region to estimate the velocity time series from 2018 to 2020.

## 3. Results

### 3.1 Collapse basin surface elevation change

After the basin surface rose by up to 38 m during 2012-2014 (Willis et al., 2015), the elevation of the entire basin continued to increase during the ArcticDEM period (2012-2017) (Figure 2a). The surface of the main and thumb basins uplifted by up to 65 m and 50 m, respectively, while the southern part of the collapse basin only had a maximum uplift of ~10 m. Figures 2b-e show sequential elevation profiles for four reference tracks across the basin. Over the main basin, profiles AA' and BB' demonstrate that a rapid surface rise of ~20 m occurred and the shape of the basin surface changed between May 2012 and March 2013. After that, the surface elevation increased more gradually by another ~40 m during 2013-2019. The elevation reached its peak value of ~660 m in April 2019, which is just ~25 m lower than the pre-collapse surface derived from AW3D30 DEM (the thick red solid line in Figure 2b-e). The ice surface elevation then showed a sudden decrease in 2019, followed by a gradual increase since January 2020. Profiles CC' and DD' show that the elevation changed gradually while the surface maintained approximately the same shape.

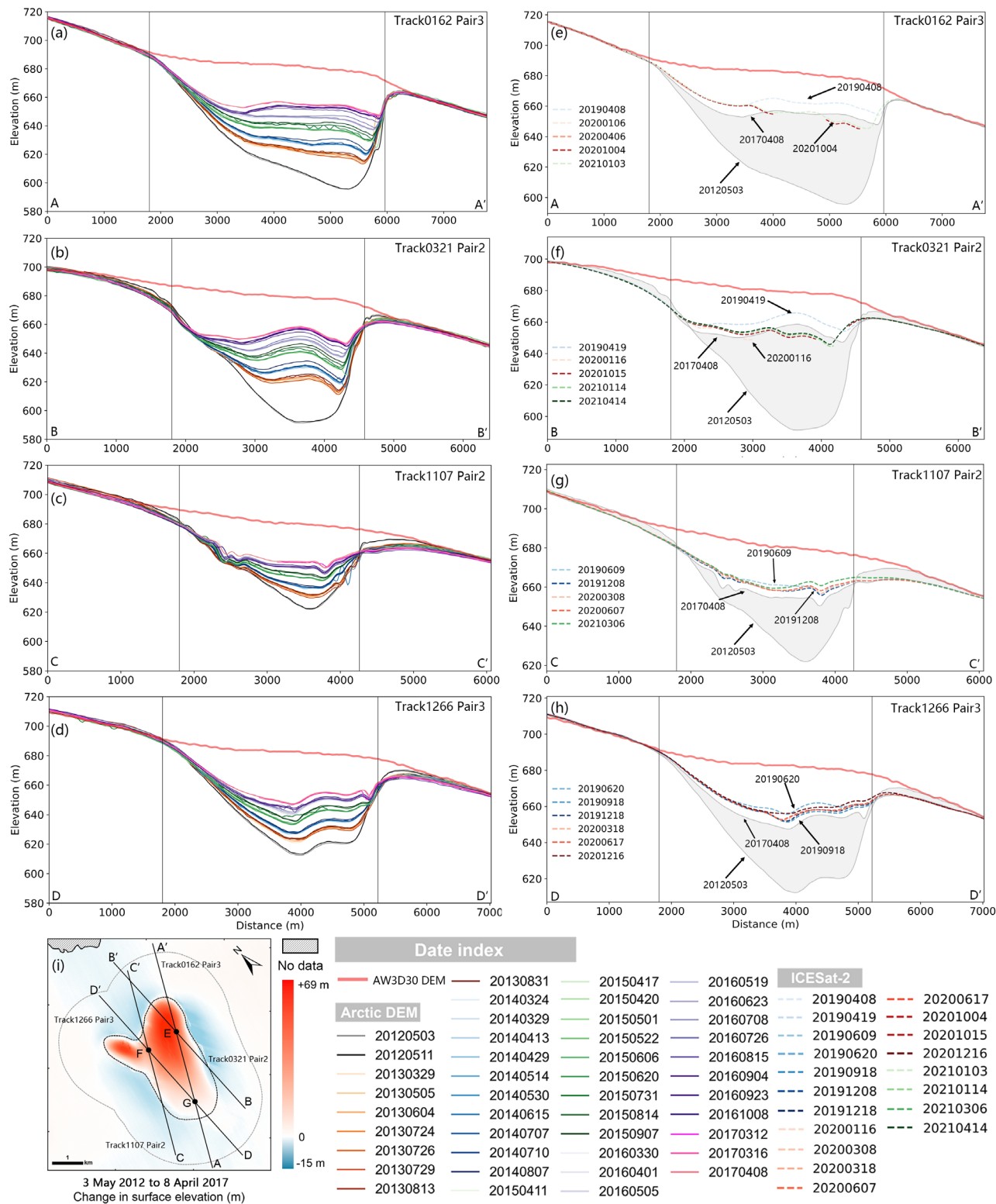

**Figure 2. Surface elevation changes from 2012 to 2021. (a-d) Repeat elevation profiles derived from ArcticDEM data from 2012 to 2017. The start and end of the profile AA', BB', CC' and DD' are shown in (i). The thick red solid line represents the elevation profile derived from AW3D30 DEM which has a timestamp of 2006–2010. The vertical lines demonstrate the position of the collapse basin boundary. (e-h) Same as (a-d) but derived from ICESat-2 data acquired between 2019 and 2021. The light gray area indicates the range of steady surface uplift between 2012 and 2017. (i) Change in surface elevation between 5 May 2012 and 8 April 2017 (DEM20170408-DEM20120505). The solid lines show the position of the reference track used to extract the elevation profiles. The black dashed curve is the boundary of the collapse basin which has an area of about 7.6 km². The gray dashed curve demonstrates the 1500-m buffer area that used to calculate the ice inflow volume. The map projection is polar stereographic (EPSG: 3413).**

Combining ArcticDEM and ICESat-2 data, we estimate changes in surface elevation at three crossovers (Figure 3). Elevation at the south edge of the collapse basin (crossover G) continuously increased by ~10 m from 2012 to 2021. At the shallow saddle between the main basin and the thumb basin (crossover F), the surface rose at a faster rate of ~5 m/yr during 2012-2021, with a sudden subsidence of ~2 m between 20 June and 18 September in 2019. The main basin (crossover E) had the most rapid surface uplift of ~9 m/yr from May 2012 to April 2019. After continuously increasing for the 8-years after the basin first collapsed in 2011, the surface of the main basin subsided by more than 10 m between 19 April 2019 and 16 January 2020. Afterward, the elevation increased again at a rate of ~5 m/yr. The elevation increased dramatically in the melt season during 2014-2016 (Figure 3 inset). During the melt season in 2014 and 2015, the surface of the main basin rose ~3 m at a rate of ~33 m/yr and ~28 m/yr, respectively. In 2016, the elevation gained ~7 m between 8 July and 4 September. This rate of elevation increase of ~49 m/yr is about half of the observed rapid surface uplift during the two-week period in 2012 (Willis et al., 2015).

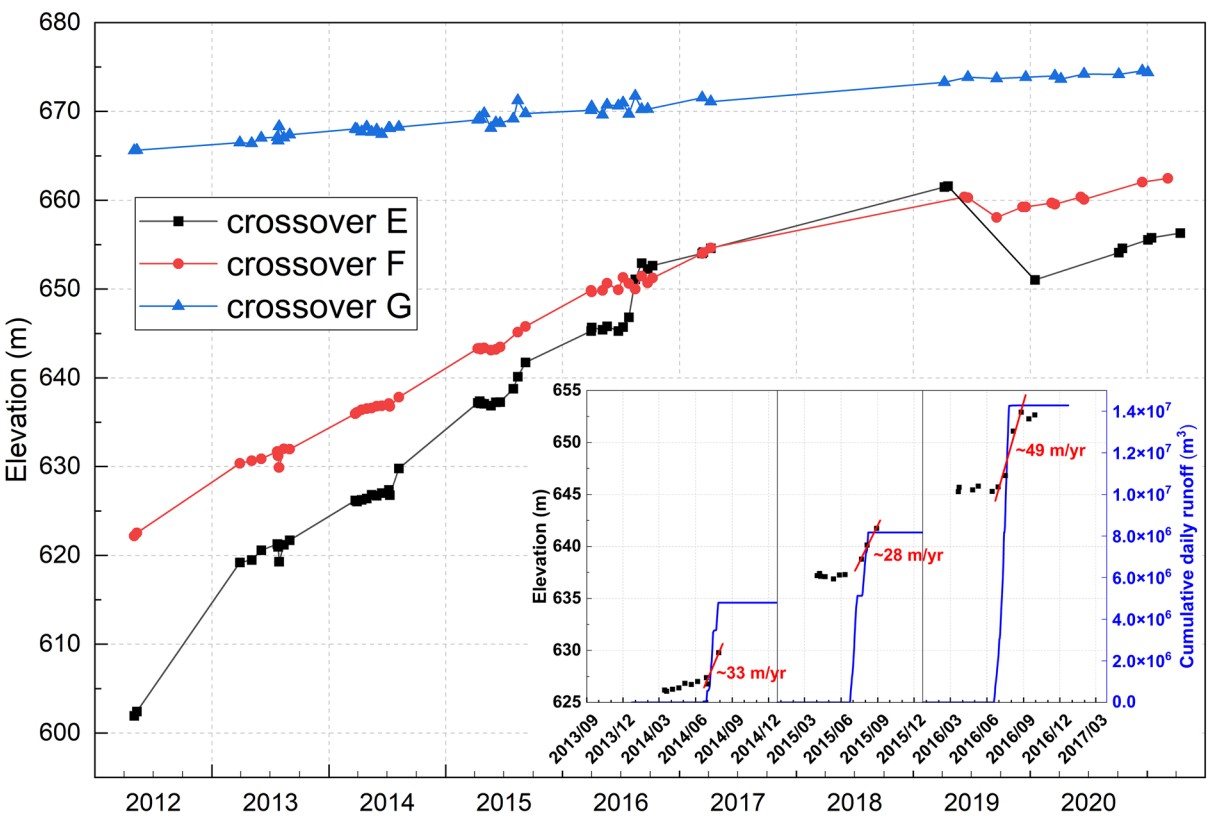

Figure 3. Ice surface elevation change from 2012 to 2021 at the three ICESat-2 crossovers shown in Fig. 2a. Crossover E demonstrates elevation change at the main basin. Crossover F demonstrates elevation change at the shallow saddle between the main basin and the thumb basin. Crossover G demonstrates elevation change at the south edge of the collapse basin. Inset shows elevation changes during 2014-2016 at Crossover E. Red lines are the average rate of increase during the period of rapid uplift each year. The blue lines show the cumulative catchment runoff from RACMO2.3p2 model.

## 3.2 Subglacial lake volume change and surface meltwater runoff

We define the volume of the collapse basin to be the volume between the pre-collapse ice surface and the post-collapse ice surface. Time series of volume change of the collapse basin and subglacial lake during the period of 2012-2017 are shown in Figure 4. Between 3 May 2012 and 5 May 2013, the volume of the collapse basin decreased by $47.5 \times 10^6$ m³, with ~55% ($26.3 \times 10^6$ m³) of the changes as a result of surface uplift caused by increasing subglacial lake volume, with the remainder due to rapid infilling by ice flow. Since 2013, however, the rate of ice inflow slowed, accounting for a small portion of the basin volume loss.

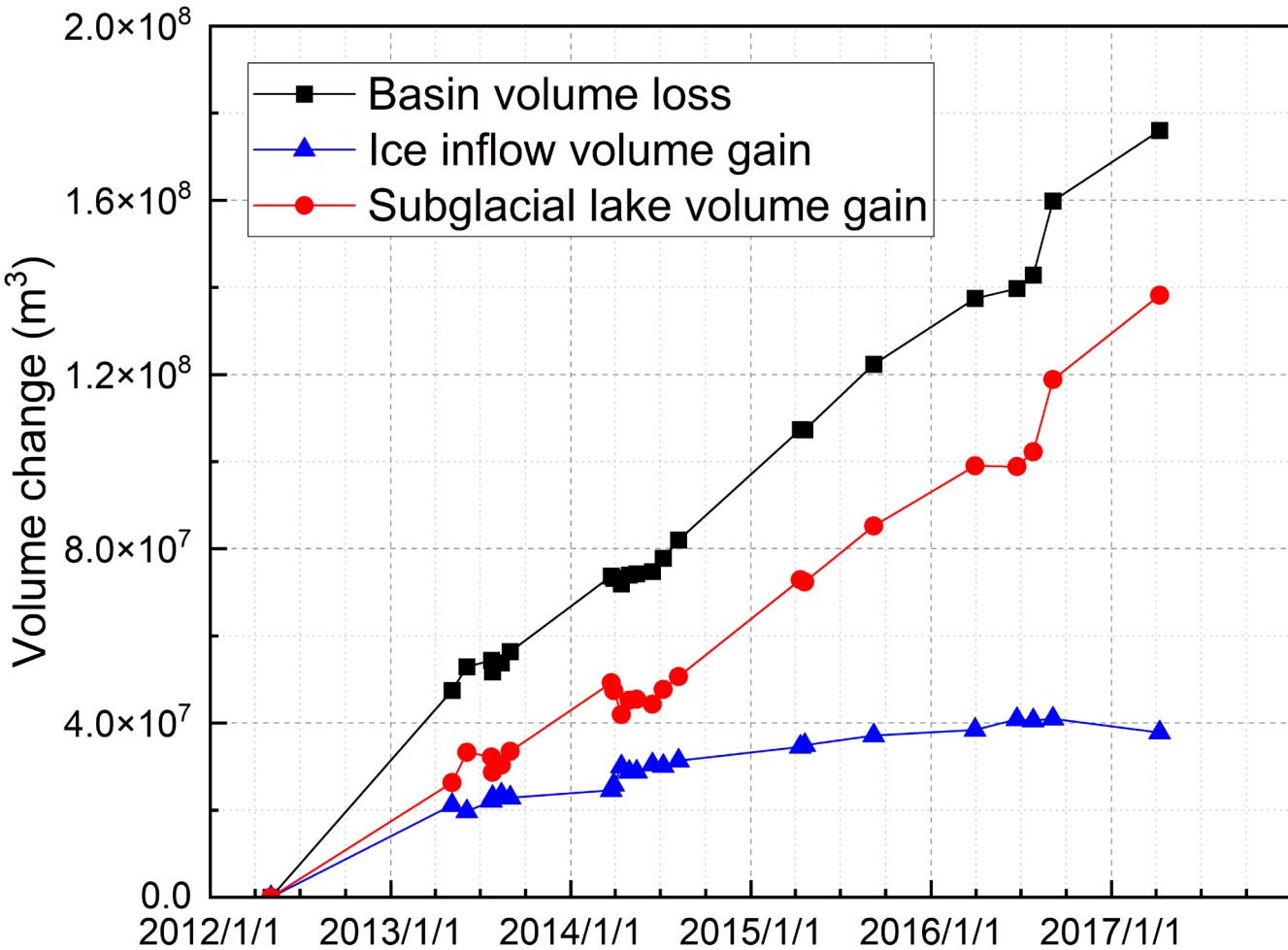

**Figure 4. Volume change of the collapse basin, ice inflow and the subglacial lake relative to 3 May 2012. Volume gain of the subglacial lake, which is caused by influx of surface meltwater, is derived by differencing the basin volume loss and ice inflow volume gain. DEMs with large voids in the buffer area were discarded to avoid potential biases.**

Basin volume showed notable changes corresponding with rapid surface uplift in the 2014-16 melt seasons. In the 2014 melt season, the basin lost a total volume of $4.2 \times 10^6$ m$^3$ between 7 July and 7 August, with the majority of the loss ($3.0 \times 10^6$ m$^3$) due to influx of surface meltwater to the subglacial lake. During the 2016 melt season, the volume of the surface basin decreased by $17.0 \times 10^6$ m$^3$ between 26 July and 4 September, and ~97% ($16.6 \times 10^6$ m$^3$) of the volume change was due to subglacial lake refilling. Over the entire 5-year period, the collapse basin lost $176.0 \times 10^6$ m$^3$ of volume. About ~21% ($37.8 \times 10^6$ m$^3$) of the loss was due to ice inflow and the remaining, $138.2 \times 10^6$ m$^3$ was the result of subglacial lake refilling by surface meltwater.

## 4. Discussion

Few active subglacial lakes have been observed in Greenland (Bowling et al., 2019; Howat et al., 2015; Livingstone et al., 2019; Palmer et al., 2015; Willis et al., 2015). This may be partly because subglacial lakes under the GrIS are nearly eight times smaller than in Antarctica (Bowling et al., 2019) and, therefore, may not be resolved by altimetry observations due to sparse track density at a relatively low polar latitude. Alternatively, the surface of the GrIS margin is typically steeper than in Antarctica, making the depressions in hydraulic potential required for lake formation less likely to occur (Howat et al., 2015).

In addition, efficient subglacial drainage systems formed in the melt season may release the stored water, preventing subglacial lake formation. Here we investigate an active subglacial lake located under the Flade Isblink ice cap is on the periphery of, and separated from, the northern GrIS. Despite this, the lake is similar in size to the active lakes found beneath the ablation zone of the GrIS.

   Willis et al. (2015) first discovered the sudden subglacial lake drainage event under Flade Isblink ice cap during the autumn

of 2011. As a result, a collapse basin was formed on the surface of the ice cap and the surface rose over the next two years due to recharging of the subglacial lake. Our estimates of the collapse basin and subglacial lake volume change between 3 May 2012 and 5 May 2013 are in agreement with Willis et al. (2015), who reported a similar amount of volume change of $46.5\times10^6$ m$^3$ and $29.6\times10^6$ m$^3$, respectively. Additionally, we also concur that volume change caused by ice inflow accounted for a large portion of basin volume loss over the first two years (2012-2014) of our investigation period. The rate

of influx declined as the depression became shallower in the following years, decreasing its contribution to basin volume change.

   Surface meltwater may drain into crevasses or moulins every melt season and lead to a rapid elevation increase in a short period. Contrasting with the north-flowing meltwater that mainly drained into crevasses on the southern margin of the collapse basin in 2012 (the polygon in Figure 5a), much of the meltwater accumulated locally in a supraglacial lake at the

southern part of the basin during the 2014-2016 melt season (Figure 5c). Changes in supraglacial hydrology may have been due to the burial of the crevasses and the significant remaining surface relief (Figure 5a&b). Following the switch in drainage location from the basin-edge crevasses in 2012 to moulins within the basin during 2014-2016, the rate of surface meltwater drainage decreased. This is confirmed by the decreasing rate of basin surface elevation uplift during the melt season. From the time of surface meltwater draining into moulins and the observed rapid uplift of the main basin during

2014-2016, we conclude that surface meltwater recharged the subglacial lake every melt season. Moreover, the larger amount of meltwater observed in 2016 corresponded to larger elevation gains. All of these processes indicate that the subglacial lake volume is primarily controlled by supraglacial meltwater filling.

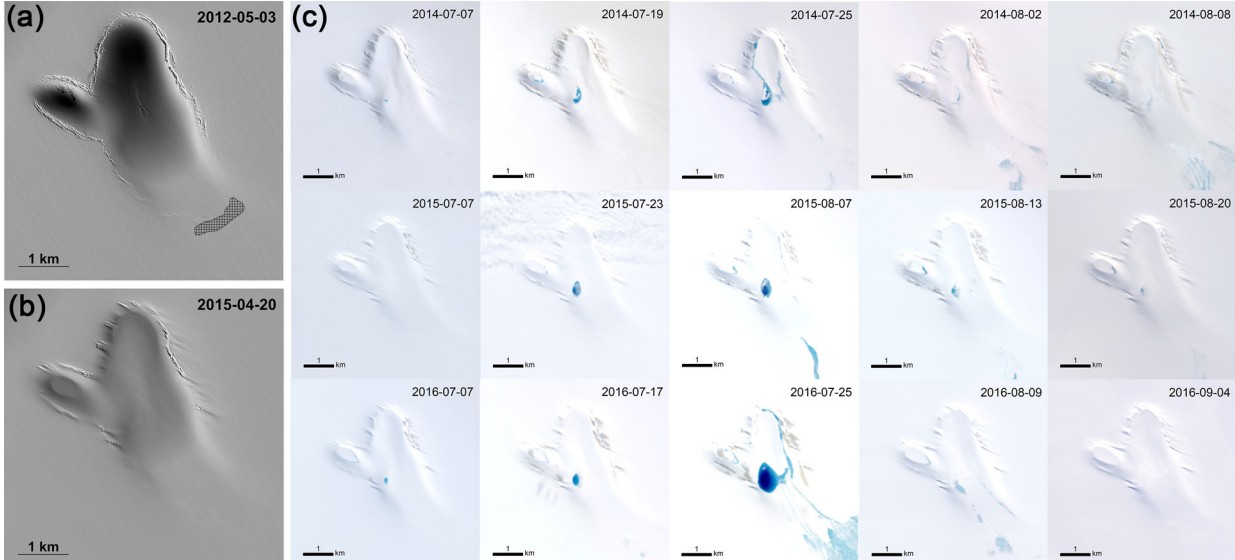

Figure 5. (a) Shaded relief images of ArcticDEM over the collapse basin in 2012. Polygon filled with diagonal crosses indicates the areas of crevasses where surface meltwater drained in 2012. (b) Shaded relief images obtained from DEMs in 2015. (c) Sequence of Landsat-8 optical imagery showing the surface meltwater evolution during 2014-2016 melt season. Each column from left to right represents different stages of melting.

Between 19 April 2019 and 16 January 2020, the surface of the main basin lowered by more than 10 m (Figure 3). We conclude this surface lowering is most likely due to drainage of the subglacial lake, which is further confirmed by Sentinel-2 images acquired at the end of August 2019 (Figure 6). Between 24 and 26 August 2019, obvious surface lowering is observed over the main basin and a distinct depression formed at the thumb basin area (Figures 6a-b), indicating a rapid subglacial lake drainage event occurred during this time. Lacking elevation measurements at the main basin in 2019 prevents us from estimating the exact duration of drainage events. While according to the elevation variation at the shallow saddle between the main basin and the thumb basin (crossover F), we speculate the drainage may end in September. The time and duration of this drainage event is consistent with previous large subglacial lake drainage events identified in Greenland (Howat et al., 2015; Livingstone et al., 2019; Palmer et al., 2015; Willis et al., 2015), which usually initiated at a time when subglacial drainage system becomes efficient and meltwater drains through connected channels (Howat et al., 2015). However, the volume of water drained in the 2019 event would be much less than in 2011, indicating that a large amount of meltwater remained in the subglacial lake. This partial subglacial lake drainage process is rare in Greenland, but have been observed beneath ice caps in Iceland where the subglacial lakes may become sealed before draining all the water (Björnsson, 2003).

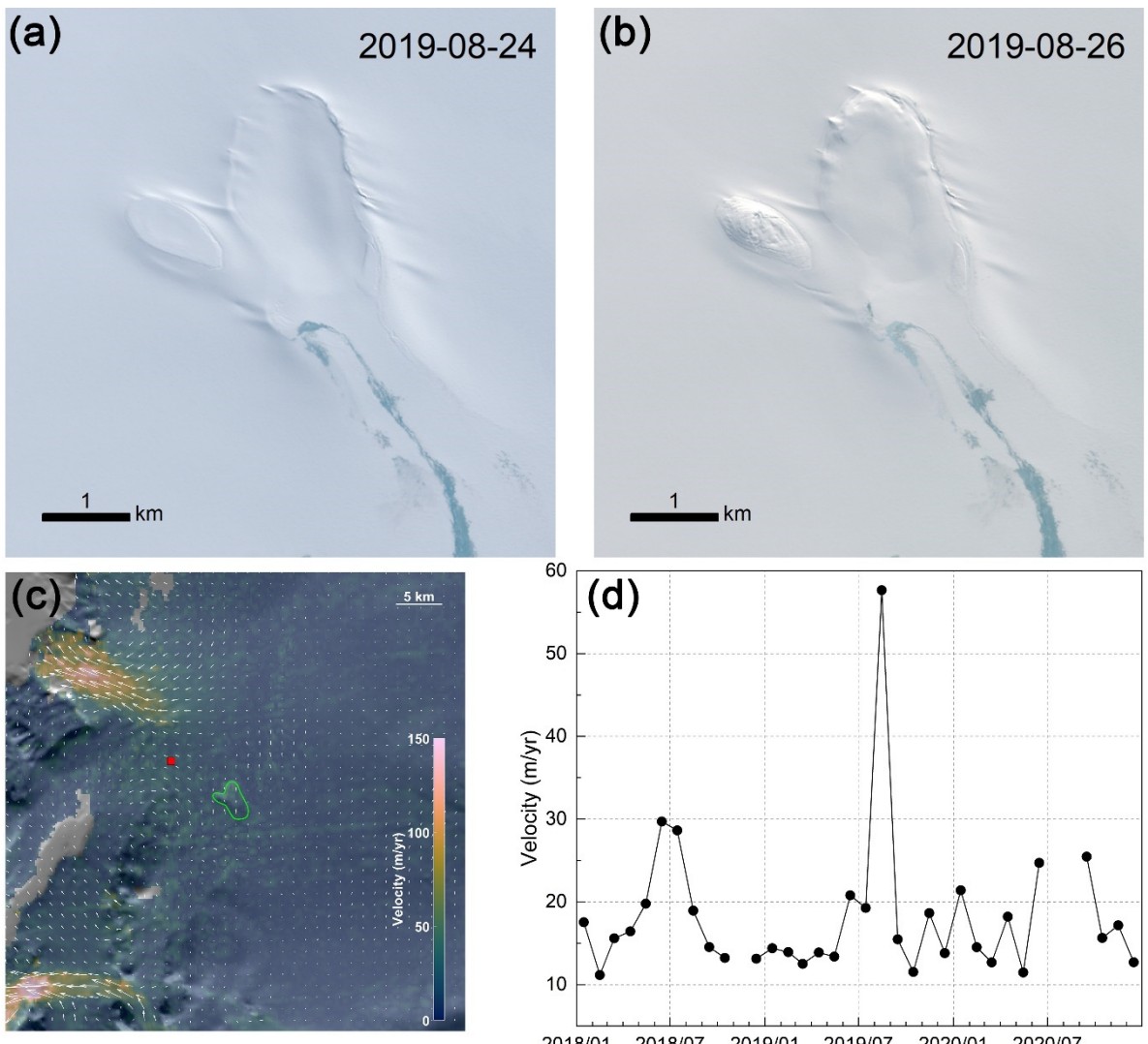

**Figure 6. Sentinel-2 optical imagery of the collapse basin and ice surface velocity around it. (a) and (b) are the images showing the obvious surface lowering between 24 and 26 August 2019. (c) The velocity map of September 2019 overlain on MODIS Mosaic of Greenland (MOG) 2015 image maps (Haran et al., 2018). The red square indicates the region of velocity averaging for the velocity time series shown in (d). The green polygon represents the boundary of the collapse basin. The white velocity vectors show direction and magnitude of horizontal velocity. (d) The velocity time series between 2018 and 2020. Each dot represents a monthly average velocity derived from MEaSUREs dataset. Note that data gaps exist due to lack of valid data in that month.**

In addition, variations in ice flow speed are consistent with water pressure variations expected during subglacial lake drainage. In August 2019, ice flow immediately downglacier from the basin increased by a factor of three over the pre-subsidence values (Figure 6d) before decreasing back to average values in the following month. We conclude that these abrupt changes resulted from the drainage event, as meltwater released from the subglacial lake initially overwhelmed the drainage system, resulting in a larger increase in water pressure and sliding speed. As the subglacial drainage system

increased in efficiency and/or the discharge of water decreased as the meltwater was drained, water pressures and sliding speeds declined.

Continued basin surface uplift from 2011 to 2019 suggests that the subglacial lake was not filled by supraglacial meltwater in a single melt season and that water storage persisted after the initial lake collapse basin initially formed. We speculate that the subglacial lake may be located upstream of a topographic ridge that would form a depression in the hydropotential field and, therefore, could store meltwater draining from the surface (Howat et al., 2015; Palmer et al., 2015). As meltwater is stored, the piezometric head within the lake increases until it exceeds the hydropotential gradient holding it in place, causing discharge. Once discharge begins, melting of channel walls at high water pressures would cause rapid expansion of the drainage system, increase in efficiency, and drainage until the piezometric head in the lake lowers, and discharge decreases and then ceases. Accumulation of meltwater draining from the surface then begins until another subglacial drainage event occurs.

Additionally, the elevation profiles through the collapse basin (Figure 2) indicate that the subglacial lake may have not been fully filled when the drainage event occurred in 2019. Based on the above findings, we speculate that this subglacial lake exhibits a pattern of slow filling and rapid drainage, similar to all active lakes beneath the Icelandic ice caps (Livingstone et al., 2022). In contrast, three active lakes beneath the GrIS are characterized by long periods of quiescence (Livingstone et al., 2019). However, similar to the Flade Isblink lake studied here, drainage of those three active lakes is not associated with high surface melt years and the duration of the drainage event is less than one month. This implies that the timings and behaviors of repeat filling and drainage of Greenland's subglacial lakes are not only determined by water storage volume, but also by meltwater input variability (Schoof, 2010) and bedrock relief (Bowling et al., 2019).

Subglacial lake water in Greenland is sourced either from geothermal and frictional melting or surface meltwater input (Bowling et al., 2019). The temperature at the bed of Flade Isblink ice cap is far below the pressure melting temperature and the ice moves relatively slowly, ruling out the local production of basal meltwater (Willis et al., 2015). Moreover, the Flade Isblink ice cap is isolated from the GrIS, hence the subglacial lake is not connected to the subglacial hydrology network beneath the GrIS. Therefore, surface meltwater is likely the only supply for this subglacial lake. Supraglacial meltwater would be routed to the bed through crevasses and moulins and flow towards the ice margin, inducing ice flow variations. A modelling study has estimated that, during an average melt season, about 39% and 47% of the surface runoff are drained through crevasses and moulins in west Greenland, respectively (Koziol et al., 2017). However, only a portion of this surface meltwater would access the ice bed interface (Nienow et al., 2017). Our results show that $3.0 \times 10^6$ m$^3$ of supraglacial water reached the subglacial lake over a one month period (7 July to 7 August) during the 2014 melt season. At the same time, total surface runoff produced within the catchment is estimated to be $4.7 \times 10^6$ m$^3$. Thus, only ~64% of the surface meltwater successfully descended to the bed. The remainder may be refrozen locally in the underlying snowpack (Harper et al., 2012). As firn aquifers and ice slabs exist around the collapse basin area (MacFerrin et al., 2019; Miller et al., 2022), part of the meltwater may also store in the firn aquifers (Forster et al., 2014; Kuipers Munneke et al., 2014) or be restricted to flow

within the firn above ice slabs (MacFerrin et al., 2019). However, we also cannot rule out the possibility of other drainage paths, subglacial or supraglacial, that we have not resolved.

## 5. Conclusion

In the autumn of 2011, a collapse basin about 70 m deep formed in the surface of the Flade Isblink ice cap in northern Greenland due to sudden subglacial lake drainage. Using multi-temporal ArcticDEM and ICESat-2 altimetry data, we document changes in surface elevation of the lake basin and estimate the subglacial lake volume change from 2012 to 2021. The long-term measurements imply that the subglacial lake was most likely recharged by seasonal influx of surface meltwater. The surface of the collapse basin rose by up to 55 m over the 9 years, with $138.2 \times 10^6$ m$^3$ of meltwater transported to the subglacial lake during 2012-2017. Our findings on the Flade Isblink ice cap that the subglacial lake can store meltwater over multiple years and decrease runoff to the ice margin are helpful for better understanding the hydrological processes on the GrIS. During our investigation period, a second rapid drainage event occurred in late August 2019, resulting in an abrupt ice velocity change. Compared to the 2011 drainage event, the amount of water drained in 2019 is much smaller and was likely only a portion of the stored water, suggesting partial drainage. In addition, the 2019 drainage was not associated with high surface melt years. These suggest that the triggering of subglacial lake drainage and subsequent evolution may be controlled by multiple factors and needs further investigation. Furthermore, a model of surface melt over the catchment estimates that only ~64% of the surface meltwater successfully descended to the bed, implying the importance of quantifying the routing of surface meltwater inputs to the ice bed interface. We have also shown that the new ICESat-2 data has great potential in detecting and monitoring active subglacial lakes beneath the GrIS.

*Data availability.* ArcticDEM can be obtained from the Polar Geospatial Center (https://www.pgc.umn.edu/data/arcticdem/). ICESat-2 ATL06 data can be obtained from National Snow and Ice Data Center (NSIDC) (https://nsidc.org/data/atl06). MEaSUREs Greenland Monthly Ice Sheet Velocity Mosaics can be obtained from NSIDC (https://nsidc.org/data/NSIDC-0731/versions/3). Landsat-8 images can be obtained from the United States Geological Survey (USGS) (https://earthexplorer.usgs.gov/). Sentinel-2 images can be obtained from the European Space Agency (https://scihub.copernicus.eu/dhus/#/home). AW3D30 DEM can be obtained from the Japan Aerospace Exploration Agency (JAXA) (https://www.eorc.jaxa.jp/ALOS/en/dataset/aw3d30/aw3d30_e.htm). RACMO2.3p2 Greenland daily runoff data were kindly provided by Brice Noël.

*Author contributions.* QL and LZ conceived the study. WX and QL processed the data and QL wrote the manuscript. All authors contributed to the data analysis and result interpretation.

*Competing interests.* The authors declare they have no conflict of interest.

*Acknowledgements.* This research was supported by the National Science Foundation for Distinguished Young Scholars (41925027) and the Innovation Group Project of Southern Marine Science and Engineering Guangdong Laboratory (Zhuhai) (No. 311021008).

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
