# Peer review of "Filling and drainage of a subglacial lake beneath the Flade Isblink ice cap, northeast Greenland"

_The Cryosphere, 2021_

## Author Comment (AC2)

Response to RC 2:
(The reviewer comments appear in back, the responses are in blue and the proposed changes to manuscript are in **_bold italics_**.)

The manuscript presents a time-series analysis from 2012-2021 of ice surface elevation change over a subglacial lake beneath the Flade Isblink ice cap. This analysis is a continuation of the initial study and identification of this subglacial lake by Willis et al. (2015), where the lake was identified via a collapsed ice surface indicating rapid lake drainage. Here, surface elevation data from the ArcticDEM and IceSat-2 is used to infer refilling and drainage of the subglacial lake. Ice surface elevation changes are compared to surface meltwater runoff from RACMO to infer the relative amount of surface meltwater input into the subglacial lake. Finally, a speedup of ice surface velocity downstream of the subglacial lake is linked to lake drainage in 2019.

**General comments**
This is well structured and generally well written manuscript that presents a new and extensive dataset revealing the filling/drainage cycle of a subglacial lake. The data is well presented and the scientific quality of the work is strong. The methods are mostly clear, however, could benefit from a few clarifications. While the manuscript does not represent novel concepts or methods itself, I believe that the observations can help address important scientific questions regarding subglacial lakes, and thus fits well within the scope of The Cryosphere. However, I find that a lot of details in the data/observations are presented, but not thoroughly discussed (for example the switch in surface meltwater drainage pattern after 2012 or the smaller lake drainage volumes in 2019 compared to 2011). This leaves me wondering about the "so what" question, and I believe that the manuscript could put a bit more emphasis on discussing the implications of the findings rather than mostly only presenting the data. Below are some specific points that I believe can be addressed with some minor revisions to improve the manuscript.
We thank the reviewer for detailed comments and suggestions.

- **Ice inflow**: The concept of surface elevation change due to ice inflow and how this was calculated is not clear. I understand that this is explained in Willis et al, 2015, but I think it would help the reader understand if is briefly explained here.

Thank you for pointing this out. We will add some text to explain more about the "inflow into the basin".
The text added/changed here will be:
"**_Previous studies show that a reduction in the depth of the depression would result from the infow of the ice around the basin (Aðalgeirsdóttir et al., 2000; Willis et al., 2015). Therefore, we expect that the basin volume change here is mainly caused by ice inflow and subglacial lake filling. Assuming the subsidence that occurs around the basin outline in a 1500-m buffer region correspond to ice flowing into the basin, we calculate the inflow volume by integrating the surface elevation changes over the buffer area (Willis et al., 2015)._**"

- **Ice surface velocity change**: It is unclear why and how this specific small area was chosen to evaluate changes in ice surface velocity. Is this based on where subglacial water routing is expected (e.g. from water routing models)? I believe that it would be better to include a larger area in the analysis, or present velocity time series from multiple locations downstream. Another idea would be to present an additional map (rather than time series) with the velocity difference between January 2019 and July 2019 to infer the velocity changes in the wider region.

Thank you for the suggestion. We do understand that more velocity variations analysis is desirable, and we would very much like to show the velocity change map or velocity change profile. But unfortunately, the coverage of the velocity product is poor in July and August 2019, due to few successful matching pixels in the summer when intense surface melting usually happened. Therefore, we are limited in where we can sample the velocity and choose to calculate the mean velocity from a region (800m*800m) located downstream of the collapse basin.

We will modify the text to clarify why this specific small area was chosen to evaluate changes in ice surface velocity.

"*Due to the limited coverage of the ice velocity product in the summer, a 800m\*800m region that is located downstream of the collapse basin is chosen to evaluate changes in ice surface velocity (Figure 6c). We calculate the mean velocity within this region to estimate the velocity time series from 2018 to 2020.*"

- **Figure 2**: I generally like Figure 2 as one can clearly see the surface elevation rising from 2012-2019. However, the surface lowering in 2019 and uplift afterwards is difficult to see. I suggest separating this time period (2019-2021) into a different graph, maybe in another four subplots to the right? If this way the panels become too small, I suggest putting graph a) and the legend to the bottom of the plot.

Thank you for the suggestion. We will improve this figure accordingly.

- **Discussion of surface meltwater drainage change**: It would be great to add the location of the meltwater drainage through the crevasses from 2012 to Figure 5, so that the changes can be observed more clearly. If possible, I suggest marking the location of these crevasses on one of the panels in Figure 5 or adding a separate panel from 2012. I am also curious of why there is a change in supraglacial hydrology (e.g. changes in surface slope?), and how the different drainage locations (crevasses at the edge of basin versus drainage through moulin within the basin) would affect the subglacial lake and basin volume changes. I feel that this change in meltwater routing is presented, but then not fully discussed.

Thank you for the suggestion. We will mark the location of these crevasses in Figure 5. In the revised version, we will also try to discuss the reason of supraglacial hydrology changes and how these changes affect the subglacial lake and basin volume changes during 2014-2016.

- **Discussion of lake drainage 2011 vs. 2019**: The lake drainage in 2019 is briefly

discussed, however, I think that there could be a bit more discussion on the difference in water release between 2011 and 2019. For example, the possibility that the lake is behind a bedrock ridge is mentioned (L238-240), but why would there be a release of all water in 2011 and not in 2019? And are there other observations of partial lake drainage elsewhere? Similarly, it would be interesting to compare the volume/time of water increase/drainage to other subglacial lakes, e.g. using the inventory by Livingstone et al, (2022). And finally, what implications could the remaining water in the subglacial lake have? E.g. would we expect another lake drainage in a few years, and would this cause a speedup or potentially a GLOF?

Livingstone, S. J., Li, Y., Rutishauser, A., Sanderson, R. J., Winter, K., Mikucki, J. A., et al. (2022). Subglacial lakes and their changing role in a warming climate. Nature Reviews Earth & Environment, 1–19. https://doi.org/10.1038/s43017-021-00246-9

Thank you for the suggestion. We agree that the initial version did not sufficiently discuss the lake drainage in 2019. In the revised version, we will add a comparison on the water drainage time and volume, and give more discussion about the partial lake drainage and the future evolution of the subglacial lake.

- **Language/grammar**: The manuscript is mostly clear and concisely written, however, there are a few instances where the grammar/language would benefit from some minor editing. I've added a few suggestions in the specific comments, but I probably didn't catch everything.

We regret there were problems with the English. In the revised version, the manuscript will be carefully checked by a native English speaker to make sure the grammar is correct.

**Specific comments**

L20: I suggest replacing "e.g." with "such as"
Agreed. We will correct this.

L24: I suggest changing to "..,which need to be further quantified"
Agreed. We will correct this.

L38-43, 185: I suggest adding a link to the most recent subglacial lake inventory:
Livingstone, S. J., Li, Y., Rutishauser, A., Sanderson, R. J., Winter, K., Mikucki, J. A., et al. (2022). Subglacial lakes and their changing role in a warming climate. Nature Reviews Earth & Environment, 1–19. https://doi.org/10.1038/s43017-021-00246-9
Agreed. We will cite this inventory in the proper place.

L45: Previously the abbreviation GrIS is used with "the" GrIS, I suggest making this consistent throughout the text.
Thank you for catching this. We will use "the GrIS" throughout the text for consistency.

L50-54: The figure caption misses a few articles, e.g. "Background is a Landsat-8…", "The black box shows the location of b"
Thank you for catching this. We will correct this.

L53: I believe that "Blue lines" be replaced with "Black lines" in the text (Figure 2b).
Thank you for catching this. We will correct this.

L58: I suggest changing to "as supraglacial meltwater was transported to the ice base, refilling the subglacial lake."
Agreed. We will correct this.

L59: The sentence structure is a bit misleading; the similar glacial setting of the Flade Isblink ice cap subglacial lake to the GrIS is probably not the "main reason" to study this lake. But studying the Flade Isblink subglacial lake can lead to important improvements in our understanding of subglacial lakes beneath the GrIS. I suggest changing the sentence structure to be more clear.
Agreed. We will modify the sentence to improve clarity.
"*Although this subglacial lake is located under the ice cap which is not directly connected to the wider GrIS, its glacial setting is similar to that of GrIS. It is important to investigate its behavior and influence, as it can lead to improvements in our understanding of subglacial lakes beneath the GrIS.*"

L75: I suggest outlining the 1500 m buffer zone for the ArcticDEM co-registration to Figure 1b, so that is more clear where this zone is.
Thank you for the suggestion. We will show this buffer zone in Figure 2.

L84: I suggest deleting "accurate"
Agreed. We will correct this.

L90: add "…(4 pairs) that pass through…"
Agreed. We will correct this.

L91: I suggest changing "pass" to "passing"
Agreed. We will correct this.

L97-98: It is not clear to me what is meant by "original elevations", please specify.
"original elevations" here means the elevation value directly extracted from the ICESat-2 data. We will modify this sentence for clarify.
"*… and comparing to the original elevations that extracted from the ICESat-2 data for the two adjacent measurements.*"

L110: It is not entirely clear how the elevation change due to ice inflow is derived. I think adding a brief section to explain the concept and how this was calculated would help the reader better understand.

Agreed. We will add some text to explain more about the "inflow into the basin".
The text added/changed here will be:
"***Previous studies show that a reduction in the depth of the depression would result from the infow of the ice around the basin (Aðalgeirsdóttir et al., 2000; Willis et al., 2015). Therefore, we expect that the basin volume change here is mainly caused by ice inflow and subglacial lake filling. Assuming the subsidence that occurs around the basin outline in a 1500-m buffer region correspond to ice flowing into the basin, we calculate the inflow volume by integrating the surface elevation changes over the buffer area (Willis et al., 2015).***"

L120: add "… runoff within the catchment…"
Thank you for catching this. We will correct this.

L123: add "acquired during the 2014-202…"
Thank you for catching this. We will correct this.

L145: change to $km^2$
Thank you for catching this. We will correct this.

L156: change to …"at a rate of …"
Thank you for catching this. We will correct this.

L169: It is not entirely clear to me what the volume of the collapse basin contains; Is it the volume between the pre-collapse ice surface and the post-collapse (and rising) ice surface, e.g. filled with air? Or is it the combination of the subglacial lake water and the ice column above? It might be good to clarify this. From the explanation of "decreasing basin volume", I assume it is the basin volume filled with air. It might also be good to then specify on Figure 4 that the Basin volume change is a volume loss, whereas the ice flow and subglacial lake volume change is a volume gain.
Thank you for the suggestion. Indeed, here the volume of the collapse basin means the volume between the pre-collapse ice surface and the post-collapse ice surface. We will add a sentence to clarify this and modify the description of the volume change in Figure 4.
"***We define the volume of the collapse basin to be the volume between the pre-collapse ice surface and the post-collapse ice surface.***"

L200: It would be great to show the drainage pattern in 2012 as compared to 2014-16.
Thank you for the suggestion. We will mark the location of the crevasses where meltwater mainly drained in 2012 in Figure 5.

L255-257: I appreciate the speculation about the "missing" surface meltwater, but is there any evidence for firn aquifers or ice slabs in this area? From a quick check, it looks like there are some ice slabs marked on the Flade Isblink ice cap by MacFerrin et al. (2019) (dataset here:

https://figshare.com/articles/dataset/Greenland_Ice_Slabs_Data/8309777), but it could be worth checking with the exact subglacial lake coordinates.

MacFerrin, M., Machguth, H., As, D. van, Charalampidis, C., Stevens, C. M., Heilig, A., et al. (2019). Rapid expansion of Greenland's low-permeability ice slabs. Nature, 573(7774), 403–407. https://doi.org/10.1038/s41586-019-1550-3

Alternatively, could surface meltwater be routed to the bed through moulins/crevasses at other locations, and then flow somewhere else and not into the subglacial lake? Could other supraglacial lake drainage routes to the bed be observed on satellite imagery?

Thank you for the suggestion. We carefully check the data and find that the ice slabs exist around the collapse basin area, though not exactly under the collapse basin. We agree with the reviewer that cannot rule out the possibility that meltwater flow somewhere else and not into the subglacial lake. While we include the entire catchment area to calculate the total runoff, so no other supraglacial lake would drain into this lake. We will modify text to clarify this.

"*As firn aquifers and ice slabs exist around the collapse basin area (MacFerrin et al., 2019; Miller et al., 2022), part of the meltwater may also stored in the firn aquifers (Forster et al., 2014; Kuipers Munneke et al., 2014) or be restricted to flow within the firn above ice slabs (MacFerrin et al., 2019). Moreover, we cannot rule out the possibility that the surface meltwater flow somewhere else and not into the subglacial lake.*"

L271: This last sentence seems a bit blunt and out of context. I suggest rephrasing to emphasize that the new satellite data has great potential in detecting and monitoring active subglacial lakes beneath the GrIS.

Thank you for the suggestion. We will modify this sentence to

"*We have also shown that the new ICESat-2 data has great potential in detecting and monitoring active subglacial lakes beneath the GrIS.*"

---

## Author Response (AR1)

Author's point-to-point response on Referee Comment #1 to TC-2021-374
(The reviewer comments appear in back, the responses are in blue and the proposed changes to manuscript are in ***bold italics***. L# refers to that in the track-changes file.)

Liang and others report the filling and drainage of a collapse basin at the Flade Isblink ice cap in northeast Greenland. The study builds on the investigations by Willis et al. (2015), who show that the basin was formed in 2011 by sudden subglacial lake drainages. Liang and others extend the time series of the observations made by Willis et al. (2015) with surface elevation estimates on the basis of digital elevation models of ArcticDEM and ALOS data as well as ICESat-2 laser altimetry. They show that the ice surface of the collapsed basin rose by 55 m after the drainage event in 2011. The authors link the ice surface uplift with a refilling of the subglacial lake and ice flowing into the basin. Furthermore, they correlate the amount of water needed for the refilling with the amount of surface meltwater produced at the ice surface.

In summary, I believe that this manuscript is an important contribution to the literature on subglacial water activity of the Greenland Ice Sheet and Greenlands smaller ice caps and fits into the scope of The Cryosphere. The manuscript is overall clearly written, and the results are well presented, however, some of the statements made need some clarification. Overall, I believe that the manuscripts, with some modifications and clarifications, merits publishing.

We thank the reviewer for a positive assessment.

1. General comments

The scientific quality of the work presented is generally strong, with good associated analysis and discussion. The methodology is almost completely inspired by the Willis et al. (2015) paper, which is probably not very innovative but has the advantage that it is an appropriate continuation of the observations made at this ice cap. Furthermore, the methods are in most parts clearly described, which enable others to reproduce the results. The figures are of good quality and appropriate. The paper structure is generally easy to follow and is mostly clearly written and clear in its conclusions.

Thank you for this review. We agree with all the suggestions and believe that these will improve the quality of the paper.

Main points:

1) Introduction

The introduction provides useful information about the study region and subglacial lakes in Greenland. However, I think a paragraph describing the versatile methods, which have been used to detect subglacial lake activity (even though most of them are located in Antarctica) should also be introduced here to give the reader an impression of what is available and what you are using in your study. In addition to laser altimetry and DEMs it would be useful to complement these two methods with:

- SAR tracking e.g., Joghin et al. (2016); https://doi.org/10.1002/2016GL070259
- InSAR: Neckel et al. (2021); https://doi.org/10.1029/2021GL094472 and Gray et al. (2005); https://doi.org/10.1029/2004GL021387

Thank you for the suggestion. In the revised version, we added a paragraph describing the methods of detecting subglacial lake activity.
L50-56:
**"Satellite remote sensing techniques have been used to monitor the subglacial lakes and detect their activities. As an indirect observation of subglacial lake activity, long-term ice surface elevation changes are usually derived from satellite altimetry (e.g., Fricker et al., 2007; Siegfried and Fricker, 2018, 2021). More recently, time-stamped digital elevation models (DEMs) have been utilized to reveal the detailed patterns of surface deformation (e.g., Livingstone et al., 2019; Willis et al., 2015). A few studies also use the Synthetic Aperture Radar (SAR) speckle tracking (Joughin et al., 2016; Hoffman et al., 2020) and Interferometry SAR (InSAR) (Gray et al., 2005; Neckel et al., 2021) to detect ice surface displacements."**

2) Figure 2
This is a great figure and shows that the authors have put a lot of thought into how they present their results. However, there is a lot of information and a lot of lines with all colours of the colour spectrum. I have a suggestion on how the results you describe maybe a little bit easier to visualize:

- What about if subfigures (b-e) are split into two columns, where you show in the left column the same style of figure, but only showing the ice surface rise until 2019. In the second column, you can show then the ice surface subsidence and the following uplift in 2020.
- For the plots in the right column, you could still show the lines from the uplift before 2019 in slight grey or so in the background. This might be a way to disentangle the two processes you describe: the steady uplift and the sudden drainage in 2019 and continuous filling thereafter.

Thank you for the suggestion. We improved this figure accordingly.

3) Throughout the manuscript, not enough attention is paid to the fact that the Flade Isblink ice cap is not directly connected to the rest of the Greenland Ice Sheet. Here it should be ensured that this is generally separated and argued in such a way that the results and interpretations of this study, however, can of course be applied to the Greenland Ice Sheet and are thus definitely very helpful.

Agreed. In the revised version, we added several sentences to clarify the disconnection between the subglacial lake and the GrIS. We also gave more discussion on the difference/similarity between the ice sheet subglacial lake and the ice cap subglacial lake in the discussion section.
L219-221:
**"Here we investigate an active subglacial lake located under the Flade Isblink ice cap is on the periphery of, and separated from, the northern GrIS. Despite this, the lake is similar in size to the active lakes found beneath the ablation zone of the GrIS."**
L288-293:
**"Based on the above findings, we speculate that this subglacial lake exhibits a pattern of slow filling and rapid drainage, similar to all active lakes beneath the Icelandic ice**

*caps (Livingstone et al., 2022). In contrast, three active lakes beneath the GrIS are characterized by long periods of quiescence (Livingstone et al., 2019). However, similar to the Flade Isblink lake studied here, drainage of those three active lakes is not associated with high surface melt years and the duration of the drainage event is less than one month.*"

L301-302:

"*Moreover, the Flade Isblink ice cap is isolated from the GrIS, hence the subglacial lake is not connected to the subglacial hydrology network beneath the GrIS.*"

L321-323:

"*Our findings on the Flade Isblink ice cap that the subglacial lake can store meltwater over multiple years and decrease runoff to the ice margin are helpful for better understanding the hydrological processes on the GrIS.*"

4) Ice inflow into the basin

One thing in the manuscript that I found difficult to understand was the idea that "volume change is mainly caused by ice inflow into the basin" (L109). I finally understood this, when I read the paper of Whillis et al. (2015) where it is explained a little bit better. So that means that the ice inflow that you talk about in your paper is related to the subsidence that occurs around the basin outline in your 1.5m buffer as indicated in Figure 2a. And the most likely interpretation is that this negative volume flows into the basin (because where else should it go). If this is what you mean, I have the following recommendations:

- Please explain a little bit better what you mean by "inflow into the basin" and that this concept is taken from the Whillis et al. (2015) manuscript.
- I think it would also be helpful to (for example) plot the 1.5 km buffer in Figure 2a and also visualize with an arrow that all the blue area (the subsidence) is the volume that flows into the basin.

Thank you for pointing this out. We added/changed some text to explain more about the "inflow into the basin". And the 1.5 km buffer area were shown in Figure 2a.

L123-128:

"*Previous studies show that a reduction in the depth of the depression would result from the inflow of the ice around the basin (Aðalgeirsdóttir et al., 2000; Willis et al., 2015). Therefore, we expect that the basin volume change here is mainly caused by ice inflow and subglacial lake filling. Assuming the subsidence that occurs around the basin outline in a 1500-m buffer region correspond to ice flowing into the basin, we calculate the inflow volume by integrating the surface elevation changes over the buffer area (Willis et al., 2015).*"

5) Treatment of ice surface flow velocity analysis after the lake drainage (e.g.

L127-129 and Figure 6c,d): I think the analysis of ice velocity variations after the drainage event should be done along a flow line instead of a single point location. At the moment we can just see that "somewhere" downstream of the lake the velocity increased, which makes me very curious what happened up- and downstream of this location (and more importantly when!).

Thank you for pointing this out. We do understand that more velocity variations analysis is desirable, and we would very much like to show the velocity change map or velocity change profile. But unfortunately, the coverage of the velocity product is poor in July and August 2019, due to few successful matching pixels in the summer when intense surface melting usually happened. Therefore, we are limited in where we can sample the velocity and choose to calculate the mean velocity from a region (800m*800m) located downstream of the collapse basin.

We modified the text to clarify why this specific small area was chosen to evaluate changes in ice surface velocity.

L144-147:

"***Due to the limited coverage of the ice velocity product in the summer, an 800 m by 800 m region located downstream of the collapse basin is chosen to evaluate changes in ice surface velocity (Figure 6c). We calculate the mean velocity within this region to estimate the velocity time series from 2018 to 2020.***"

6) There is a slight inconsistent usage of the term "Greenland Ice Sheet" and its abbreviation "GrIS". Please make this consistent:

- "Greenland ice sheet" vs. "Greenland Ice Sheet"
- "GrIS" vs. "the GrIS"

Thank you for catching this. We use "Greenland Ice Sheet" and "the GrIS" throughout the text for consistency.

7) Overall the language of the manuscript is clear, but I have the impression that the grammar is not always correct. This should be checked by a native English speaker.

We regret there were problems with the English. In the revised version, the manuscript was carefully checked by a native English speaker to make sure the grammar is correct.

2. Specific comments

L38-49: Please update the paragraphs information of the current knowledge and database of subglacial lakes in Greenland by the findings of the recent review of Livingstone et al. (2022):

- Livingstone, S. J., Li, Y., Rutishauser, A., Sanderson, R. J., Winter, K., Mikucki, J. A., et al. (2022). Subglacial lakes and their changing role in a warming climate. Nature Reviews Earth & Environment, 1–19. https://doi.org/10.1038/s43017-021-00246-9

Thank you for the suggestion. We updated this.

L56: Reference "(Willis et al., 2015)" without brackets.

Thank you for catching this. We corrected this.

L57: "DEM" is not defined anywhere. Although most of the readers might know what it stands for, it would be good to define it here once.

Thank you for catching this. We defined it when it first appears.

L59: "[...] this subglacial lake is under the ice cap [...]" should be "[...] this subglacial lake is located under the ice cap [...]"

Thank you for catching this. We corrected this.

L234: You state that "The repeat filling and drainage of the subglacial lake is on the scale of ~8 years". However, then you state later that the lake was probably not completely full in the 2019 drainage event. Also when I have a look at Figure 3 I do not fully have the impression that the small drainage event in 2019 would allow speaking of a real cyclicity of filling and drainage of the lake. I agree that "something happened" in 2019, but when I compare this to the original collapse basin surface in Figure 2, I have more the feeling that the 2019 event was a small outburst, but still in the "filling process" (what you also state later in the conclusions as "partial drainage". You do later argue that "the repeat filling and drainage is not only decided by the volume of water stored in the subglacial lake but also may be controlled by meltwater input … and bedrock relief …". However, all this is still very speculative, hence, I recommend removing this statement about the "repeat filling drainage cycle of 8 years".

We agree. We avoid using the statement of "repeat filling drainage cycle of 8 years" throughout the paper.

L245-248: Here you discuss why it is most likely that "surface meltwater is likely the only supply for this subglacial lake." I think at this point it should again be made clear that the Flade Isblink ice cap is isolated from the Greenland Ice Sheet and therefore not connected to its subglacial hydrology network, which further supports the idea that supraglacial meltwater that finds its way to the ice-cap bottom is the most likely source for the lake filling.

Thank you for the suggestion. We added a sentence here to emphasize the separation.
L301-302:

"*Moreover, the Flade Isblink ice cap is isolated from the GrIS, hence the subglacial lake is not connected to the subglacial hydrology network beneath the GrIS.*"

L261: In the conclusions, you state that: "The long-term measurements show that the subglacial lake was recharged by surface meltwater produced in the melt season". I think this should be stated as the most likely scenario instead of as a fact. Your observations are good and convincing but are based on remote sensing data only and the hypothesis is mainly based on observations made in other regions in Greenland where supraglacial water reached the bed. Hence, you cannot prove that the water at this ice cap makes its way down to the bed, although you have strong arguments for it. I would recommend stating that this is "the most likely scenario" instead of that the measurements "show" it.

Agreed. We modified the text to:
L319-320:

"*The long-term measurements imply that the subglacial lake was most likely recharged by seasonal influx of surface meltwater.*"

L263-265: Furthermore you state: "Our work demonstrates the potential for subglacial lake to store multi-year meltwater in GrIS, which may affect the ice flow by preventing the transfer of meltwater to the ice sheet margin." Here, again it should be made clear that this ice cap is not connected to the rest of the GrIS. I think it would be better to state something like that your findings on the ice cap that the subglacial lake can store multiyear meltwater [...] are useful to understand the hydrological processes on the GrIS.

Thank you for the suggestion. We modified the text to:

L321-323:

"*Our findings on the Flade Isblink ice cap that the subglacial lake can store meltwater over multiple years and decrease runoff to the ice margin are helpful for better understanding the hydrological processes on the GrIS.*"

Author's point-to-point response on Referee Comment #2 to TC-2021-374
(The reviewer comments appear in back, the responses are in blue and the proposed changes to manuscript are in ***bold italics***. L# refers to that in the track-changes file.)

The manuscript presents a time-series analysis from 2012-2021 of ice surface elevation change over a subglacial lake beneath the Flade Isblink ice cap. This analysis is a continuation of the initial study and identification of this subglacial lake by Willis et al. (2015), where the lake was identified via a collapsed ice surface indicating rapid lake drainage. Here, surface elevation data from the ArcticDEM and IceSat-2 is used to infer refilling and drainage of the subglacial lake. Ice surface elevation changes are compared to surface meltwater runoff from RACMO to infer the relative amount of surface meltwater input into the subglacial lake. Finally, a speedup of ice surface velocity downstream of the subglacial lake is linked to lake drainage in 2019.

**General comments**
This is well structured and generally well written manuscript that presents a new and extensive dataset revealing the filling/drainage cycle of a subglacial lake. The data is well presented and the scientific quality of the work is strong. The methods are mostly clear, however, could benefit from a few clarifications. While the manuscript does not represent novel concepts or methods itself, I believe that the observations can help address important scientific questions regarding subglacial lakes, and thus fits well within the scope of The Cryosphere. However, I find that a lot of details in the data/observations are presented, but not thoroughly discussed (for example the switch in surface meltwater drainage pattern after 2012 or the smaller lake drainage volumes in 2019 compared to 2011). This leaves me wondering about the "so what" question, and I believe that the manuscript could put a bit more emphasis on discussing the implications of the findings rather than mostly only presenting the data. Below are some specific points that I believe can be addressed with some minor revisions to improve the manuscript.
We thank the reviewer for detailed comments and suggestions.

- **Ice inflow**: The concept of surface elevation change due to ice inflow and how this was calculated is not clear. I understand that this is explained in Willis et al, 2015, but I think it would help the reader understand if is briefly explained here.
Thank you for pointing this out. We added some text to explain more about the "inflow into the basin".
L123-128:
"***Previous studies show that a reduction in the depth of the depression would result from the inflow of the ice around the basin (Aðalgeirsdóttir et al., 2000; Willis et al., 2015). Therefore, we expect that the basin volume change here is mainly caused by ice inflow and subglacial lake filling. Assuming the subsidence that occurs around the basin outline in a 1500-m buffer region correspond to ice flowing into the basin, we calculate the inflow volume by integrating the surface elevation changes over the buffer area (Willis et al., 2015).***"

- **Ice surface velocity change**: It is unclear why and how this specific small area was chosen to evaluate changes in ice surface velocity. Is this based on where subglacial water routing is expected (e.g. from water routing models)? I believe that it would be better to include a larger area in the analysis, or present velocity time series from multiple locations downstream. Another idea would be to present an additional map (rather than time series) with the velocity difference between January 2019 and July 2019 to infer the velocity changes in the wider region.

Thank you for the suggestion. We do understand that more velocity variations analysis is desirable, and we would very much like to show the velocity change map or velocity change profile. But unfortunately, the coverage of the velocity product is poor in July and August 2019, due to few successful matching pixels in the summer when intense surface melting usually happened. Therefore, we are limited in where we can sample the velocity and choose to calculate the mean velocity from a region (800m*800m) located downstream of the collapse basin.

We modified the text to clarify why this specific small area was chosen to evaluate changes in ice surface velocity.

L144-147:

"***Due to the limited coverage of the ice velocity product in the summer, an 800 m by 800 m region located downstream of the collapse basin is chosen to evaluate changes in ice surface velocity (Figure 6c). We calculate the mean velocity within this region to estimate the velocity time series from 2018 to 2020.***"

- **Figure 2**: I generally like Figure 2 as one can clearly see the surface elevation rising from 2012-2019. However, the surface lowering in 2019 and uplift afterwards is difficult to see. I suggest separating this time period (2019-2021) into a different graph, maybe in another four subplots to the right? If this way the panels become too small, I suggest putting graph a) and the legend to the bottom of the plot.

Thank you for the suggestion. We improved this figure accordingly.

- **Discussion of surface meltwater drainage change**: It would be great to add the location of the meltwater drainage through the crevasses from 2012 to Figure 5, so that the changes can be observed more clearly. If possible, I suggest marking the location of these crevasses on one of the panels in Figure 5 or adding a separate panel from 2012. I am also curious of why there is a change in supraglacial hydrology (e.g. changes in surface slope?), and how the different drainage locations (crevasses at the edge of basin versus drainage through moulin within the basin) would affect the subglacial lake and basin volume changes. I feel that this change in meltwater routing is presented, but then not fully discussed.

Thank you for the suggestion. We marked the location of these crevasses in Figure 5. In the revised version, we also tried to discuss the reason of supraglacial hydrology changes and how these changes affect the subglacial lake and basin volume changes during 2014-2016.

L233-237:

*"Changes in supraglacial hydrology may have been due to the burial of the crevasses and the significant remaining surface relief (Figure 5a&b). Following the switch in drainage location from the basin-edge crevasses in 2012 to moulins within the basin during 2014-2016, the rate of surface meltwater drainage decreased. This is confirmed by the decreasing rate of basin surface elevation uplift during the melt season."*

- **Discussion of lake drainage 2011 vs. 2019**: The lake drainage in 2019 is briefly discussed, however, I think that there could be a bit more discussion on the difference in water release between 2011 and 2019. For example, the possibility that the lake is behind a bedrock ridge is mentioned (L238-240), but why would there be a release of all water in 2011 and not in 2019? And are there other observations of partial lake drainage elsewhere? Similarly, it would be interesting to compare the volume/time of water increase/drainage to other subglacial lakes, e.g. using the inventory by Livingstone et al, (2022). And finally, what implications could the remaining water in the subglacial lake have? E.g. would we expect another lake drainage in a few years, and would this cause a speedup or potentially a GLOF?
  Livingstone, S. J., Li, Y., Rutishauser, A., Sanderson, R. J., Winter, K., Mikucki, J. A., et al. (2022). Subglacial lakes and their changing role in a warming climate. Nature Reviews Earth & Environment, 1–19. https://doi.org/10.1038/s43017-021-00246-9

Thank you for the suggestion. We agree that the initial version did not sufficiently discuss the lake drainage in 2019. In the revised version, we added a comparison on the water drainage time and volume, and give more discussion about the partial lake drainage and the future evolution of the subglacial lake.

L258-260:

*"This partial subglacial lake drainage process is rare in Greenland, but have been observed beneath ice caps in Iceland where the subglacial lakes may become sealed before draining all the water (Björnsson, 2003)."*

L281-285:

*"As meltwater is stored, the piezometric head within the lake increases until it exceeds the hydropotential gradient holding it in place, causing discharge. Once discharge begins, melting of channel walls at high water pressures would cause rapid expansion of the drainage system, increase in efficiency, and drainage until the piezometric head in the lake lowers, and discharge decreases and then ceases. Accumulation of meltwater draining from the surface then begins until another subglacial drainage event occurs."*

L288-293:

*"Based on the above findings, we speculate that this subglacial lake exhibits a pattern of slow filling and rapid drainage, similar to all active lakes beneath the Icelandic ice caps (Livingstone et al., 2022). In contrast, three active lakes beneath the GrIS are characterized by long periods of quiescence (Livingstone et al., 2019). However, similar to the Flade Isblink lake studied here, drainage of those three active lakes is*

*not associated with high surface melt years and the duration of the drainage event is less than one month.*"

- **Language/grammar**: The manuscript is mostly clear and concisely written, however, there are a few instances where the grammar/language would benefit from some minor editing. I've added a few suggestions in the specific comments, but I probably didn't catch everything.

We regret there were problems with the English. In the revised version, the manuscript was carefully checked by a native English speaker to make sure the grammar is correct.

**Specific comments**

L20: I suggest replacing "e.g." with "such as"
Agreed. We corrected this.

L24: I suggest changing to "..,which need to be further quantified"
Agreed. We corrected this.

L38-43, 185: I suggest adding a link to the most recent subglacial lake inventory: Livingstone, S. J., Li, Y., Rutishauser, A., Sanderson, R. J., Winter, K., Mikucki, J. A., et al. (2022). Subglacial lakes and their changing role in a warming climate. Nature Reviews Earth & Environment, 1–19. https://doi.org/10.1038/s43017-021-00246-9
Agreed. We cited this inventory in the proper place.

L45: Previously the abbreviation GrIS is used with "the" GrIS, I suggest making this consistent throughout the text.
Thank you for catching this. We use "the GrIS" throughout the text for consistency.

L50-54: The figure caption misses a few articles, e.g. "Background is a Landsat-8…", "The black box shows the location of b"
Thank you for catching this. We corrected this.

L53: I believe that "Blue lines" be replaced with "Black lines" in the text (Figure 2b).
Thank you for catching this. We corrected this.

L58: I suggest changing to "as supraglacial meltwater was transported to the ice base, refilling the subglacial lake."
Agreed. We corrected this.

L59: The sentence structure is a bit misleading; the similar glacial setting of the Flade Isblink ice cap subglacial lake to the GrIS is probably not the "main reason" to study this lake. But studying the Flade Isblink subglacial lake can lead to important improvements in our understanding of subglacial lakes beneath the GrIS. I suggest changing the sentence structure to be more clear.
Agreed. We modified the sentence to improve clarity.

L68-71:
"*Although Flade Isblink ice cap is not directly connected to the wider GrIS, its glacial setting is similar to that of the northern GrIS. It is important to investigate its behavior and impact on ice dynamics, which may lead to improvements in our understanding of subglacial lakes beneath the GrIS.*"

L75: I suggest outlining the 1500 m buffer zone for the ArcticDEM co-registration to Figure 1b, so that is more clear where this zone is.
Thank you for the suggestion. We showed this buffer zone in Figure 2.

L84: I suggest deleting "accurate"
Agreed. We corrected this.

L90: add "…(4 pairs) that pass through…"
Agreed. We corrected this.

L91: I suggest changing "pass" to "passing"
Agreed. We corrected this.

L97-98: It is not clear to me what is meant by "original elevations", please specify.
"original elevations" here means the elevation value directly extracted from the ICESat-2 data. We modified this sentence for clarify.
L109-111:
"*Further, we check for height consistency by calculating adjacent elevations using the along-track slope parameter and comparing the estimated to the measured elevations.*"

L110: It is not entirely clear how the elevation change due to ice inflow is derived. I think adding a brief section to explain the concept and how this was calculated would help the reader better understand.
Agreed. We added some text to explain more about the "inflow into the basin". Please refer to our reply to the first general comments for details.

L120: add "… runoff within the catchment…"
Thank you for catching this. We corrected this.

L123: add "acquired during the 2014-202…"
Thank you for catching this. We corrected this.

L145: change to km$^2$
Thank you for catching this. We corrected this.

L156: change to …"at a rate of …"
Thank you for catching this. We corrected this.

L169: It is not entirely clear to me what the volume of the collapse basin contains; Is it the volume between the pre-collapse ice surface and the post-collapse (and rising) ice surface, e.g. filled with air? Or is it the combination of the subglacial lake water and the ice column above? It might be good to clarify this. From the explanation of "decreasing basin volume", I assume it is the basin volume filled with air. It might also be good to then specify on Figure 4 that the Basin volume change is a volume loss, whereas the ice flow and subglacial lake volume change is a volume gain.

Thank you for the suggestion. Indeed, here the volume of the collapse basin means the volume between the pre-collapse ice surface and the post-collapse ice surface. We added a sentence to clarify this and modified the description of the volume change in Figure 4.

L194-195:

"*We define the volume of the collapse basin to be the volume between the pre-collapse ice surface and the post-collapse ice surface.*"

L200: It would be great to show the drainage pattern in 2012 as compared to 2014-16.

Thank you for the suggestion. We marked the location of the crevasses where meltwater mainly drained in 2012 in Figure 5.

L255-257: I appreciate the speculation about the "missing" surface meltwater, but is there any evidence for firn aquifers or ice slabs in this area? From a quick check, it looks like there are some ice slabs marked on the Flade Isblink ice cap by MacFerrin et al. (2019) (dataset here: https://figshare.com/articles/dataset/Greenland_Ice_Slabs_Data/8309777), but it could be worth checking with the exact subglacial lake coordinates.

MacFerrin, M., Machguth, H., As, D. van, Charalampidis, C., Stevens, C. M., Heilig, A., et al. (2019). Rapid expansion of Greenland's low-permeability ice slabs. Nature, 573(7774), 403–407. https://doi.org/10.1038/s41586-019-1550-3

Alternatively, could surface meltwater be routed to the bed through moulins/crevasses at other locations, and then flow somewhere else and not into the subglacial lake? Could other supraglacial lake drainage routes to the bed be observed on satellite imagery?

Thank you for the suggestion. We carefully checked the data and found that the ice slabs exist around the collapse basin area, though not exactly under the collapse basin. We agreed with the reviewer that cannot rule out the possibility that meltwater flow somewhere else and not into the subglacial lake. While we included the entire catchment area to calculate the total runoff, so no other supraglacial lake would drain into this lake. We modified text to clarify this.

L310-314:

"*As firn aquifers and ice slabs exist around the collapse basin area (MacFerrin et al., 2019; Miller et al., 2022), part of the meltwater may also store in the firn aquifers (Forster et al., 2014; Kuipers Munneke et al., 2014) or be restricted to flow within the firn above ice slabs (MacFerrin et al., 2019). However, we also cannot rule out the possibility of other drainage paths, subglacial or supraglacial, that we have not*

*resolved.*"

L271: This last sentence seems a bit blunt and out of context. I suggest rephrasing to emphasize that the new satellite data has great potential in detecting and monitoring active subglacial lakes beneath the GrIS.
Thank you for the suggestion. We modified this sentence to
L331-332:
"*We have also shown that the new ICESat-2 data has great potential in detecting and monitoring active subglacial lakes beneath the GrIS.*"

---

## Editor Decision (ED1)

Dear authors,

Thank you for your thorough response to the reviewer's comments and the revised manuscript. I am satisfied that the changes you have incorporated address the majority of their concerns, though I note they both asked for a little revision of the language. I therefore add some suggestions below that should aid interpretation and style.

I suggest beginning the paper at paragraph 2. Paragraph 1 is largely not necessary. The final sentence could be incorporated into paragraph 2, for example:

'Sixty-four subglacial lakes have been identified beneath the GrISin Greenland from airborne radio-echo sounding (Bowling et al., 2019; Livingstone et al., 2022). Most of them are stable, showing little or no evidence of volume change or input from the surface, and are located above the Equilibrium Line Altitude (ELA) and the relatively flat, frozen-bedded ice sheet interior (Bowling et al., 2019). Only a few hydrologically active lakes that are recharged by surface meltwater have been identified from ice surface elevation change measurements (Bowling et al., 2019; Howat et al., 2015; Livingstone et al., 2019; Livingstone et al., 2022; Palmer et al., 2015; Willis et al., 2015). Compared to the more widely distributed stable subglacial lakes, active subglacial lakes are affected more directly by surface meltwater and their drainage would may significantly influence the glacier flow dynamics (Davison et al., 2020; Livingstone et al., 2019). Despite this importance, our understanding of Greenland's subglacial lakes has been primarily developed from theoretical studies or inferences from geophysical exploration due to sparsity of the limited direct observations (Davison et al., 2019). The presence and movement of meltwater at the ice bed interface significantly affect ice dynamics (Meierbachtol et al., 2013). Given the expected increases in surface meltwater production in a warming climate (Mottram et al., 2017; Sellevold and Vizcaino, 2021), it is of critical importance to understand the hydrology of the Greenland Ice Sheet (GrIS), especially the routing, storage, drainage and recharge of subglacial water'

I am not sure why all the methods have changed from past to present tense: I suggest reverting

L216: 'relatively low polar latitude' – I know what you mean, but it is confusing to have 'low latitiude' and 'polar' in the same sentence. Suggest removing the final part of the sentence, so it ends with 'sparse track density'.

L218: replace 'in addition' with 'finally,'

L220: missing 'which'

L221: remove 'despite this'

L224: 'remove 'as a result' – the collapse basin did not form as a result of Willis et al discovering it

L253: rather than 'Lacking elevation measurements' suggest 'A lack of elevation mreasurements'

L258: 'the volume of water drained in the 2019 event **was likely** much less'

L259: How do you know this process is rare, when you've told us there are so few observations? Suggest instead 'The ubiquity of partial subglacial lake drainage is unknown in Greenland, but similar processes have been observed….'

L270: remove 'in addition'

L284: 'increasing efficiency of drainage until'

L288: remove 'Additionally'

L289: remove 'based on the above findings'

L290-295 is quite confusing. Please state more simply.

L311: 'The remainder may be locally refrozen in the underlying snowpack (Harper) or in firn aquifers that have been detected around the collapse basin area (refs). Ice slabs are also likely to exist locally (ref), so meltwater may be restricted or travel via other drainage paths that our study is unable to detect.'

L322-3: this new sentence is awkward here. I suggest moving it to the final sentence of the paper.

L330: remove 'furthermore'

Thank you for your submission to The Cryosphere.

Dr Liz Bagshaw, Editor

---

## Author Response (AR2)

Dear Editor,

Thank you for your very helpful suggestions in improving this manuscript. Below are our point-by-point responses to your comments. The comments appear in back, the responses are in blue and the proposed changes to manuscript are in ***bold italics***.

Best regards
Qi Liang

Thank you for your thorough response to the reviewer's comments and the revised manuscript. I am satisfied that the changes you have incorporated address the majority of their concerns, though I note they both asked for a little revision of the language. I therefore add some suggestions below that should aid interpretation and style.

I suggest beginning the paper at paragraph 2. Paragraph 1 is largely not necessary. The final sentence could be incorporated into paragraph 2, for example:

'Sixty-four subglacial lakes have been identified beneath the GrISin Greenland from airborne radio-echo sounding (Bowling et al., 2019; Livingstone et al., 2022). Most of them are stable, showing little or no evidence of volume change or input from the surface, and are located above the Equilibrium Line Altitude (ELA) and the relatively flat, frozen-bedded ice sheet interior (Bowling et al., 2019). Only a few hydrologically active lakes that are recharged by surface meltwater have been identified from ice surface elevation change measurements (Bowling et al., 2019; Howat et al., 2015; Livingstone et al., 2019; Livingstone et al., 2022; Palmer et al., 2015; Willis et al., 2015). Compared to the more widely distributed stable subglacial lakes, active subglacial lakes are affected more directly by surface meltwater and their drainage would may significantly influence the glacier flow dynamics (Davison et al., 2020; Livingstone et al., 2019). Despite this importance, our understanding of Greenland's subglacial lakes has been primarily developed from theoretical studies or inferences from geophysical exploration due to sparsity of the limited direct observations (Davison et al., 2019). The presence and movement of meltwater at the ice bed interface significantly affect ice dynamics (Meierbachtol et al., 2013). Given the expected increases in surface meltwater production in a warming climate (Mottram et al., 2017; Sellevold and Vizcaino, 2021), it is of critical importance to understand the hydrology of the Greenland Ice Sheet (GrIS), especially the routing, storage, drainage and recharge of subglacial water'

Thanks for your suggestions. Indeed, these changes will make the text easier to follow and read. We have made modification accordingly.

I am not sure why all the methods have changed from past to present tense: I suggest reverting

Thank you for the suggestion. We have changed to the past tense.

L216: 'relatively low polar latitude' – I know what you mean, but it is confusing to have 'low latitiude' and 'polar' in the same sentence. Suggest removing the final part of the

sentence, so it ends with 'sparse track density'.
Agreed. We have removed the final part.

L218: replace 'in addition' with 'finally,'
Changed.

L220: missing 'which'
Added.

L221: remove 'despite this'
Removed.

L224: 'remove 'as a result' – the collapse basin did not form as a result of Willis et al discovering it
Thanks for catching this. We have removed it.

L253: rather than 'Lacking elevation measurements' suggest 'A lack of elevation mreasurements'
Changed.

L258: 'the volume of water drained in the 2019 event was likely much less'
Changed.

L259: How do you know this process is rare, when you've told us there are so few observations? Suggest instead 'The ubiquity of partial subglacial lake drainage is unknown in Greenland, but similar processes have been observed….'
Agreed. We have changed the text as you suggested.

L270: remove 'in addition'
Removed.

L284: 'increasing efficiency of drainage until'
Changed.

L288: remove 'Additionally'
Removed.

L289: remove 'based on the above findings'
Removed.

L290-295 is quite confusing. Please state more simply.
Thank you for catching this. To clarify we have changed these sentences to:
*"The elevation profiles through the collapse basin (Figure 2) indicate that the subglacial lake may have not been fully filled when the drainage event occurred in*

*2019. This drainage is not associated with high surface melt years and the duration of the drainage event is less than one month. We speculate that this subglacial lake exhibits a pattern of slow filling and rapid drainage, similar to all active lakes beneath the Icelandic ice caps (Livingstone et al., 2022). In contrast, three active lakes beneath the GrIS are characterized by long periods of quiescence (Livingstone et al., 2019)."*

L311: 'The remainder may be locally refrozen in the underlying snowpack (Harper) or in firn aquifers that have been detected around the collapse basin area (refs). Ice slabs are also likely to exist locally (ref), so meltwater may be restricted or travel via other drainage paths that our study is unable to detect.'
Changed.

L322-3: this new sentence is awkward here. I suggest moving it to the final sentence of the paper.
Agreed. We have changed this.

L330: remove 'furthermore'
Removed.